# Patterns of Richness and Endemism in the Gypsicolous Flora of Mexico

**Juan Pablo Ortiz-Brunel** [1], **Helga Ochoterena** [2], **Michael J. Moore** [3], **Juvenal Aragón-Parada** [1], **Joel Flores** [4], **Guadalupe Munguía-Lino** [5], **Aarón Rodríguez** [1], **María Magdalena Salinas-Rodríguez** [6] and **Hilda Flores-Olvera** [2,*]

[1] Departamento de Botánica y Zoología, Centro Universitario de Ciencias Biológicas y Agropecuarias, Universidad de Guadalajara, Zapopan 45200, Mexico; jpablo.ortiz@alumnos.udg.mx (J.P.O.-B.); juvenal.aparada@alumnos.udg.mx (J.A.-P.); aaron.rodriguez@academicos.udg.mx (A.R.)

[2] Departamento de Botánica, Instituto de Biología, Circuito Exterior s.n., Ciudad Universitaria, Ciudad de México 04510, Mexico; helga@ib.unam.mx

[3] Department of Biology, Oberlin College, Oberlin, OH 44074, USA; mmoore@oberlin.edu

[4] IPICYT/División de Ciencias Ambientales, San Luis Potosí 78216, Mexico; joel@ipicyt.edu.mx

[5] Cátedras CONACyT-Universidad de Guadalajara, Centro Universitario de Ciencias Biológicas y Agropecuarias, Universidad de Guadalajara, Zapopan 45132, Mexico; guadalupe.munguia95@academicos.udg.mx

[6] Herbario Jorge Marroquín, Facultad de Ciencia Biológicas, Universidad Autónoma de Nuevo León, Avenida Pedro de Alva, s/n, San Nicolas de los Garza 66451, Mexico; mane.salinasrdr@uanl.edu.mx

\* Correspondence: mahilda@ib.unam.mx

**Abstract:** Gypsum soils occur around the world, mainly in arid regions. These harsh environments promote unusual flora with high degrees of endemism. Mexico has extensive gypsum outcrops, but their flora has been poorly studied. However, the highest species richness and endemism are expected to be concentrated in Mexico's northern dry regions. To promote the study of this flora and its conservation, we estimate how well sampled it is, quantify species richness, identify centers of endemism, and detect which gypsum outcrops lie within federal protected natural areas (PNA). We conducted exhaustive literature and herbaria reviews to generate a database of botanical records on gypsum soils. The total species and gypsophyte richness were calculated using cell grids. Centers of endemism were identified using the corrected weighted endemism index (CWE). We mapped the gypsum outcrops within PNA polygons. The most collected sites are Cuatro Ciénegas (Coahuila) and Santo Domingo Tonalá (Oaxaca), which also had the highest total species richness. Nevertheless, gypsophyte richness was higher in Cuatro Ciénegas and Nuevo León. The CWE identified seven gypsophyte centers of endemism. Mexico hosts the most diverse gypsophile flora in the world, despite having been only partially studied and collected. The regions with the highest species richness and endemism are unprotected.

**Keywords:** biodiversity; Cuatro Ciénegas; edaphic endemism; gypsophile; gypsophyte; gypsum; plants

## 1. Introduction

Gypsum soils occur worldwide. They are especially common in arid or semiarid regions, although they occur as small outcrops in wetter areas [1–4]. These soils are composed mainly of sedimentary deposits resulting from ancient high salinity environments [3]. In North America, most gypsum deposits formed in the Paleozoic or Mesozoic eras and were later exposed by Cenozoic uplift and erosion [5]. Due to its high solubility, gypsum mixes easily with adjacent soils [6]. Desert climatic conditions allow the conservation of gypsum horizons because the low precipitation slows their erosion and intermixing with different soils and greatly reduces the accumulation of organic matter [7]. The physical characteristics of gypsum soils (gypsisols) vary depending on their origin and the

local climate [3]. Gypsum can be present as anhydrite, crystalline selenite, evaporite bedrock, secondary evaporites, or even sand dunes [6]. In Mexico, gypsum is generally found as alluvial deposits, crusts, crystals, rocks, and sand dunes in bedrocks, cliffs, hills, and plains [6,8].

Mexico has multiple gypsisol areas distributed throughout the country. According to the Servicio Geológico Mexicano [9], gypsum soils occur in Baja California, Baja California Sur, Campeche, Chihuahua, Coahuila, Colima, Durango, Guerrero, Hidalgo, Jalisco, Michoacán, Morelos, Nuevo León, Oaxaca, Puebla, San Luis Potosí, Sinaloa, Sonora, Tamaulipas, and Zacatecas states. The highest concentration occurs in the north and northeastern part of the country in the states of Chihuahua, Coahuila, Nuevo León, and San Luis Potosí [4,10,11]. These zones correspond mainly to the Chihuahuan Desert (CHIH) and the Sierra Madre Oriental (SMOr) biogeographic provinces (following Morrone et al. [12]). Gypsisols are widespread over these provinces, but the size of individual gypsum exposures ranges from several square kilometers to a few square meters [8,13]. Consequently, soil maps from Mexico do not have the necessary scale to depict all the outcrops present in the country. In Baja California and Baja California Sur, the records of gypsum are scarce [14]. Additionally, gypsum soils are reported in areas in biogeographic provinces with tropical affinity in the Pacific Lowlands (PL), the Balsas Basin (BB), and the Yucatán Peninsula. These outcrops are found in the states of Campeche [15], Colima [16,17], Guerrero [18], Jalisco [19], Oaxaca [20,21], and Puebla [22].

Plants that live on gypsum are often called gypsicolous plants. Many of these have developed physical and physiological traits that promote their survival, so much so that many of them are found only in this type of soil. Some examples are seeds with a mucilaginous coat, tolerance to high (e.g., S and Ca) and/or low (e.g., P) nutrient levels, and strong roots that break through soil crusts [7,23,24]. The species that inhabit gypsum are mostly herbs, shrubs, and rarely trees [25]. Gypsicolous plants are often grouped by their frequency of occurrence on gypsum. Following Mota et al. [26], plants found only on gypsum are gypsophytes, plants that occur mostly on gypsum are gypsoclines, and those that grow on and off gypsum are called gypsovags. The proportion of these three types of plants in a gypsum community has been found to depend on the gypsum concentration in the soil and the climatic conditions [3,27]. Additionally, high concentrations of foliar sulfur and magnesium have been shown to play an important role in the adaptation of plants to gypsum [28]. Globally, the diversity of physical characteristics of the gypsum, together with the different reliefs and climates, promotes the development of many types of vegetation in gypsum [29]. In Mexico, gypsum is reported in xerophilous scrubland [6,21,30], grasslands [31–33], dune vegetation [34], tropical deciduous forests [16,17,35], and oak and pine forests [10,22,36,37] (Figure 1).

Most gypsum communities studied in Mexico are located within the CHIH [5,8,13,34,38] and the SMOr [37,39–41]. Unfortunately, most gypsum outcrops in Mexico are either poorly explored botanically or even unexplored, especially those in the western and southern portions of the country. Floristic studies on gypsophytes in Mexico have focused on a few taxonomic groups, such as pteridophytes [42], gymnosperms [36], Amaranthaceae [43], and Euphorbiaceae [22]. Alternatively, a few studies have focused on distinct areas, such as the Cuatro Ciénegas basin [44]. Recently, the exploration and study of some areas of these interesting communities in Oaxaca have resulted in outstanding novelties such as the genus *Mixtecalia* Redonda-Mart., García-Mend. and D. Sandoval (Asteraceae) [21], and several new species within the genera *Agave* L. (Asparagaceae), *Bletia* Ruiz and Pav. (Orchidaceae), *Cephalocereus* Pfeiff. (Cactaceae), *Hechtia* Klotzsch (Bromeliaceae), and *Pinguicula* L. (Lentibulariaceae) [35,45–49].

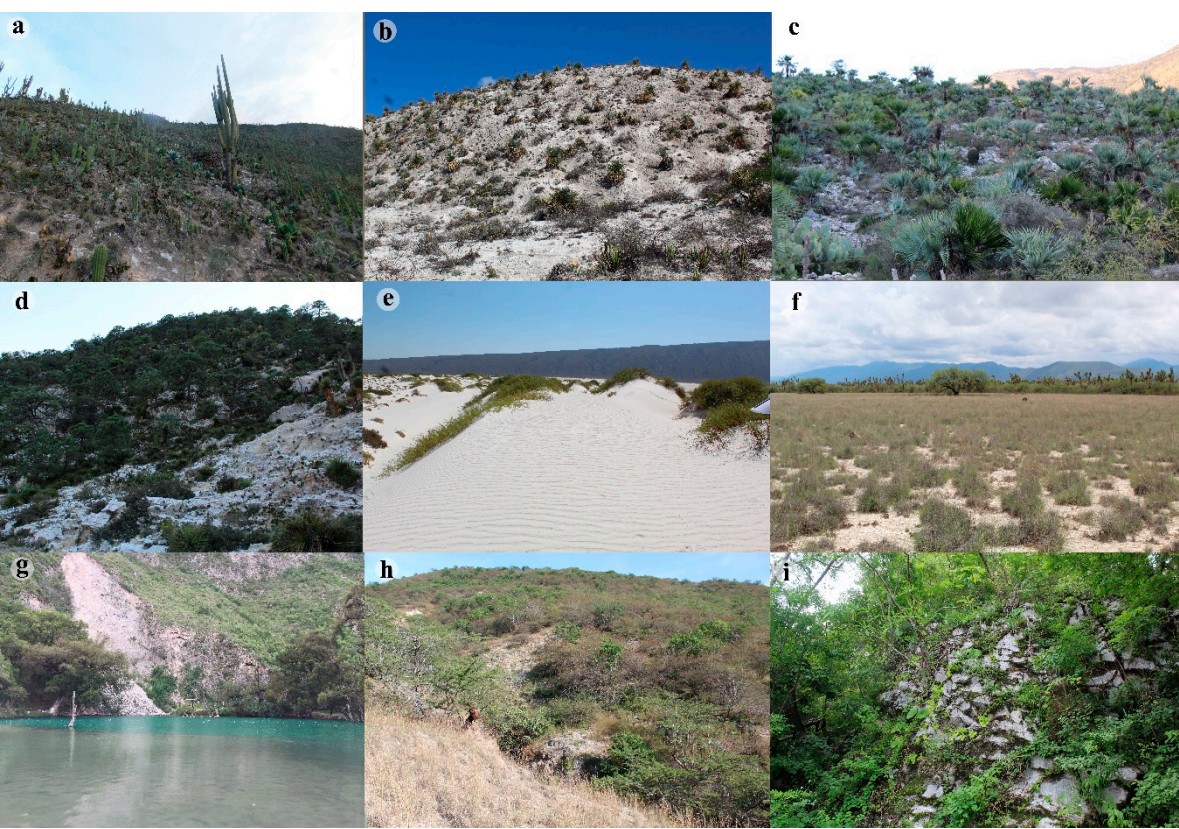

**Figure 1.** Vegetation types in gypsum soils of Mexico: (**a**) xerophilous scrubland in San Juan Teita, Oaxaca, (**b**) xerophilous scrubland in Aramberri, Nuevo León, (**c**) palmetto woodland in Galeana, Nuevo León, (**d**) coniferous forest in Galeana, Nuevo León, (**e**) dune vegetation in Cuatro Cienégas, Coahuila, (**f**) grassland in Vanegas, San Luis Potosí, (**g**) tropical deciduous forest in Santiago Juxtlahuaca, Oaxaca, (**h**) grassland and tropical deciduous forest in Tolimán, Jalisco, and (**i**) tropical deciduous forest in Tecomán, Colima. Photographs: (**a**) Juvenal Aragón, (**b–i**) J. P. Ortiz-Brunel.

Despite the works mentioned above, a great proportion of the gypsicolous flora of Mexico remains unrecorded. Moore et al. [6] estimated at least 200 gypsophyte species in the CHIH, including its portions in USA and Mexico. However, this number includes only gypsophytes and lacks gypsocline and gypsovag species. Furthermore, the number of these three kinds of gypsicolous plants throughout Mexico is unknown. Generally, gypsisol areas host an elevated number of endemic taxa [3,7]. In Mexico, Sosa and De-Nova [50] identified two hotspots of angiosperm endemism that in part comprise gypsum soils: the Northeastern rosette scrub and the Gypsum grasslands. Additionally, Salinas-Rodríguez et al. [51] identified one hotspot for the endemic vascular plants of the SMOr that includes the gypsum outcrops from Nuevo León and Tamaulipas states. Given this, it is expected that northern Mexico has the widest extent of gypsum soils and the highest species richness and endemism of gypsophytes in the country. Meanwhile, tropical gypsum outcrops from central and southern Mexico should also be expected to host a high species richness because angiosperm richness is higher in tropical and subtropical regions [52]. However, it is necessary to explicitly test these patterns with occurrence data.

To promote additional research and conservation of these ecologically and evolutionarily important but generally underappreciated habitats, this study provides the first comprehensive overview of the general state of knowledge of the gypsicolous flora in Mexico. For the first time, we identify and map the botanical records that explicitly refer to gypsum in Mexico. We also determine how well botanically sampled these sites are and estimate the spatial distribution of vascular plant species richness and centers of endemism

within them. We also assess which of these gypsum areas fall within the Protected Natural Areas (PNAs) of Mexico to detect which are currently under federal protection.

## 2. Materials and Methods

### 2.1. Botanical Records on Mexican Gypsum Soils

We conducted an exhaustive literature review of ecological and floristic studies, monographs, revisions, and species descriptions involving gypsophytes. Then, we searched for the records that included any of the following words: gipsícola, gipsófila, gypseous, gypsicolous, gypsophile, gypsophyte, gypsum, yeso, and yesoso. This allowed us to identify many gypsum areas that have botanical records in Mexico, facilitating the review of records in herbaria and electronic databases. Later, we reviewed herbarium records from ANSM, GBH, IBUG, MEXU, OAX, SERO, SLPM, TEX/LL, and ZEA (acronyms according to Thiers [53]) that included any of the words mentioned for the literature review. Additionally, we added records from the Global Biodiversity Information Facility (GBIF [54]) and IBdata [55], following the same search methodology and for which the species determination was verifiable. All the species with herbarium vouchers explicitly mentioning gypsum were included in a database and, when necessary, the localities were carefully georeferenced using Google Earth Pro 4.2. We identified the states and municipalities of Mexico to which these areas corresponded. Consequently, we aggregated records from works that explicitly used herbarium voucher references, such as the endemic vascular plants of the Sierra Madre Oriental [56], selected angiosperm groups in Mexico [57], the Cuatro Ciénegas Basin [44], and the endemic vascular plants of the Sierra Madre del Sur [58]. Gypsum is often localized in small outcrops; therefore, we excluded doubtful localities. Additionally, the records that mentioned growing on gypsum but without a herbarium voucher were not included. Several herbarium specimens were personally identified when there was conflicting identification among duplicates. To avoid bias due to different classification systems, all the species names were standardized with the Missouri Botanical Garden's Tropicos database using the Taxonomic Name Resolution Service [59]. Other taxonomical updates for higher taxonomic ranks followed APG IV [60] for angiosperms, Christenhusz et al. [61] for gymnosperms, and Christenhusz et al. [62] for pteridophytes.

From the final database, duplicated and doubtful entries were deleted using collector numbers and data filtering. All remaining records were converted into points and processed in QGIS version 2.14.3 (QGIS Development [63]). We generated a map with these points to identify the botanical records that explicitly refer to gypsum across Mexico and their biogeographic affinity following Morrone et al. [12]. We then incorporated the localities reported by the Servicio Geológico Mexicano (SGM [9]) to identify the presence of gypsum for which we were unable to find herbarium records. To estimate the intensity of botanical collections, we used the observation richness tool in DIVA-GIS 4.2 [64]. The cell size was the same as that in the species richness analyses. We are aware that the use of this approach overstates the extent of some gypsum sites. However, the purpose of this map was only to estimate the sampling status of the potential gypsicolous flora throughout Mexico according to herbarium vouchers and the literature.

### 2.2. Species Richness

The distinction between gypsovag, gypsocline, and gypsophyte is often overlooked in herbarium labels and databases; thus, we used the literature and locality comparisons to assign these categories in our database. Most species facultatively growing on gypsum in Mexico lack detailed ecological information; thus, we avoided the use of the term gypsocline. For each species, we searched all available records in digital and herbarium collections and the literature. The species that included more than two records in localities without gypsum were considered as gypsovags.

The spatial distribution of species richness was estimated by two approaches. We used both gypsophytes and gypsovags in one analysis and only gypsophytes in the other. This allowed us to identify the total species richness patterns on gypsum soils as whole

plant communities and to detect where the greatest number of gypsophytes grow all together. We performed a richness analysis by cell grid in DIVA-GIS 4.2 with a cell size of $0.18° \times 0.18°$. The cell size was obtained using the Maximum Distances method established in Willis et al. [65] and modified by Suárez-Mota and Villaseñor [66]. Even though some gypsum outcrops are considerably smaller than the cell size used, we chose this method because it is a good strategy to analyze the spatial distribution of species richness and allowed us to compare our results with other patterns of plant species richness across the country. The results from these analyses were visualized in QGIS. Finally, to test if the species richness depended on the number of observations, we applied a regression analysis between the grid results of the total species richness and those of the number of observations in DIVA-GIS.

### 2.3. Endemism

Analyzing centers of endemism is useful to identify areas for conservation [67]. Weighted endemism (WE) evaluates a cell based on the endemism level of all the species present within it. Meanwhile, the corrected weighted endemism (CWE) index adjusts the WE value of a cell as a function of its species richness [68,69]. The endemism analysis was only performed with gypsophytes because if a species is considered a gypsovag, it is assumed to have a wider distribution area, and our database does not include its complete distribution. Since the gypsophyte richness analysis shows the cells with the highest number of gypsophytes growing together, we only used WE to calculate the CWE index. This allowed us to recognize those cells with few, but very restricted, gypsophyte species. Both parameters were estimated in the program Biodiverse [70] using the same cell size as in the richness analyses.

### 2.4. Conservation Status of the Mexican Gypsicolous Flora

We used the QGIS geoprocessing tools to overlay our recorded gypsum areas with the current polygons of the Protected Natural Areas (PNAs) of Mexico as stated by the "Comisión Nacional de Áreas Naturales Protegidas" (CONANP [71]). This allowed us to identify which gypsicolous communities in Mexico are currently protected by federal law. Additionally, we identified the plant species recorded on gypsum that are protected by the Mexican Federal Law NOM-059-SEMARNAT-2010 [72].

## 3. Results

### 3.1. Botanical Records on Mexican Gypsum Soils

We identified botanical records reporting gypsum sites in 14 states of Mexico: Baja California, Baja California Sur, Chihuahua, Coahuila, Colima, Durango, Guerrero, Jalisco, Nuevo León, Oaxaca, Puebla, San Luis Potosí, Tamaulipas, and Zacatecas (Figure 2a). The states with the highest number of gypsum sites recorded were Nuevo León and Coahuila. In these two states, most outcrops occur in pine oak forests and xerophilous scrubland in the municipalities of Cuatro Ciénegas, Galeana, Aramberri, and Zaragoza. In San Luis Potosí, gypsophile grasslands are frequent in the municipalities of Matehuala and Vanegas, while xerophilous scrubland is common in Guadalcázar, San Nicolás Tolentino, and Villa Juárez. Many scattered sites were identified across Chihuahua and Coahuila, mainly in xerophilous scrubland or grassland and even in gypsum dunes, such as the ones in the Cuatro Ciénegas Basin. In Baja California and Baja California Sur, gypsum records were scarce. In Baja California there are reports between Chapala and Punta Prieta and in Baja California Sur at Isla San Marcos in Mulegé municipality. Four isolated areas were identified in Durango, while all the sites from Zacatecas were located along its borders with Coahuila or San Luis Potosí (Figure 2a). In Tamaulipas, gypsum records are concentrated in the southwestern portion of the state, on xerophilous shrubland close to Nuevo León and San Luis Potosí. In western Mexico, small gypsum outcrops occur in Colima and Jalisco, on a very restricted area of mostly limestone-derived soils with tropical deciduous forests. This also occurs in Guerrero, Puebla, and in the Santiago Juxtlahuaca and Santo

Domingo Tonalá municipalities in Oaxaca. The other sites in Oaxaca were reported as xerophilous scrubland.

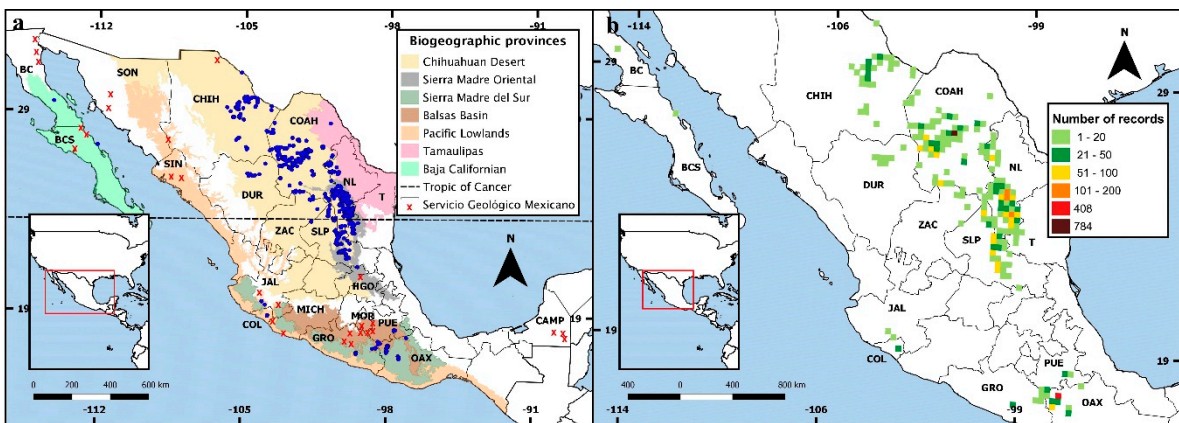

**Figure 2.** Current distribution of botanical collections on gypsum: (**a**) gypsisols with botanical records in Mexico and (**b**) number of observations in gypsum sites. The blue dots constitute all the botanical records. The red crosses represent gypsisol sites reported by the SGM [9] without botanical records. State abbreviations: BC = Baja California, BCS = Baja California Sur, CAMP = Campeche, COAH = Coahuila, COL = Colima, CHIH = Chihuahua, DUR = Durango, GRO = Guerrero, HGO = Hidalgo, JAL = Jalisco, MICH = Michoacán, MOR = Morelos, NL = Nuevo León, OAX = Oaxaca, PUE = Puebla, SIN = Sinaloa, SLP = San Luis Potosí, SON = Sonora, T = Tamaulipas, ZAC = Zacatecas.

According to the herbarium records, the CHIH province presented the highest concentration of gypsum sites, widely distributed across it (Figure 2a). However, potential gypsisols in the SMOr are abundant along its northern portion and in some cases, they were larger in extent than those in the CHIH. Additionally, there were three gypsum localities in the Tamaulipas province. Only two gypsum localities were recorded for the Baja Californian province. There were many gypsum sites recorded in biogeographic provinces with tropical affinities. In western Mexico, gypsum outcrops were documented in a reduced area pertaining to the Pacific Lowlands. Meanwhile, the gypsisols of the southern portion of the country were situated in the Balsas Basin (BB) and the Sierra Madre del Sur (SMS) provinces.

We recorded 4880 plant collections on Mexican gypsisols, but their distribution was markedly heterogeneous (Figure 2b). The Cuatro Ciénegas Basin hosted the highest sampling concentration with 784 records. Another highly sampled site, with 408 accessions, was identified in southern Mexico in the Santo Domingo Tonalá municipality in Oaxaca (Figure 2b). Finally, some areas with 51–100 and 101–200 records were localized mainly across Coahuila, Nuevo León, and San Luis Potosí. The rest of the documented sites across the country had less than 50 records, with most less than 20 (Figure 2b).

### 3.2. Species Richness

We recorded 1470 species growing on gypsum soils in Mexico. The highest total species richness (gypsophytes and gypsovags) was found in northern Mexico at the Cuatro Ciénegas Basin in Coahuila, with 288 species, and in the Santo Domingo Tonalá municipality in Oaxaca, with 266 (Figure 3a). Cells with 81–150 species were identified in the municipalities of Aramberri, Galeana, and Zaragoza in Nuevo León, and cells with 51–80 species were found in Nuevo León and San Luis Potosí. The remaining gypsum sites had less than 50 species (Figure 3a). The regression analysis showed that the variation in the total species richness pattern is strongly associated with the variation in the number of records ($R^2$ = 0.9, $p < 0.001$).

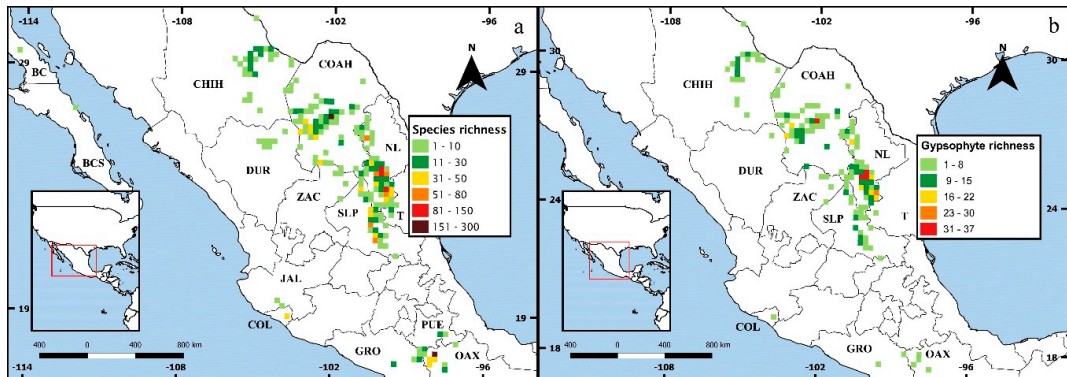

**Figure 3.** Spatial distribution of species richness within the gypsicolous flora in Mexico: (**a**) all species recorded and (**b**) gypsophytes.

Among all species, 205 were cataloged as gypsophytes (with 1878 records). The highest gypsophyte species richness was found in northern Mexico (Figure 3b). There were 37 gypsophytes recorded in the Cuatro Ciénegas Basin. Other cells with 23–36 species were found in Nuevo León in the municipalities of Galeana and Aramberri-Zaragoza. Gypsisols with 16–22 species were detected in the states of Coahuila and Nuevo Léon. All remaining sites had fewer than 15 gypsophytes, and Baja California, Baja California Sur, Jalisco, and Puebla had none (Figure 3b).

### 3.3. Endemism

The CWE analysis detected many cells across the country with high levels of local endemism (Figure 4). We identified seven important centers of endemism based on the highest valued cells and their concentration. Centers 1 and 2 were located in the CHIH, specifically in the Cuatro Ciénegas Basin, Coahuila (twelve exclusive gypsophytes), and in Mina, Nuevo León (six exclusive gypsophytes). Centers 3 and 4 occurred along the Sierra Madre Oriental. Center 3 is located in southern Nuevo León and hosts the largest number of exclusive gypsophytes (62 species), while Center 4 occurred in San Luis Potosí (eight exclusive gypsophytes). The fifth center corresponds to Colima and it is supported by two microendemic species. The last two centers were found in southern Mexico at the confluences of the BB with the SMS in Oaxaca, at the municipalities of Santo Domingo Tonalá, Santiago Juxtlahuaca, and San Juan Teita, and both sites had three microendemic gypsophytes, respectively. Table 1 shows the gypsophytes that support the proposal of centers of endemism.

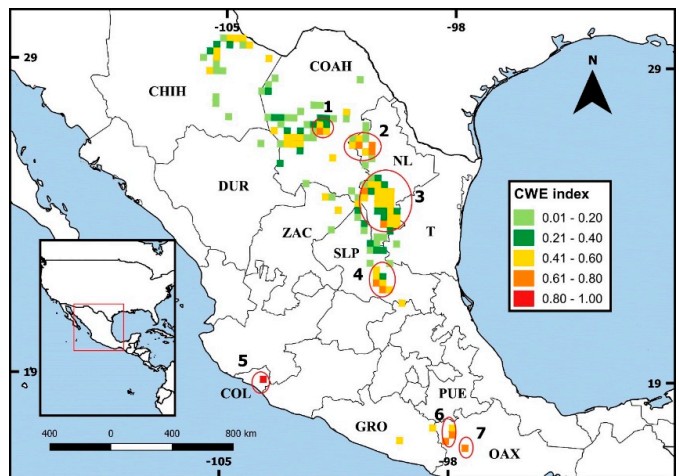

**Figure 4.** Centers of endemism detected for the gypsophytes in Mexico.

**Table 1.** Restricted species that support the gypsophyte endemism centers.

| Endemism Center | Supporting Species |
|---|---|
| 1<br>Cuatro Ciénegas Basin | Amaranthaceae: *Tidestromia rhizomatosa* I.M. Johnst.; Asteraceae: *Erigeron cuatrocienegensis* G.L. Nesom, *Gaillardia gypsophila* B.L. Turner, *Haploesthes robusta* I.M. Johnst., *Solidago gypsophila* G.L. Nesom, *Xanthisma restiforme* (B.L. Turner) D.R. Morgan and R.L. Hartm., *Xylothamia truncata* G.L. Nesom; Cactaceae: *Ancistrocactus pinkavanus* García-Mor., Gonz.-Bot. and Rodr. González; Euphorbiaceae: *Euphorbia scopulorum* Brandegee; Gentianaceae: *Sabatia tuberculata* J.E. Williams; Poaceae: *Bouteloua johnstonii* Swallen, *Sporobolus spiciformis* Swallen. |
| 2<br>Mina, Nuevo León | Asteraceae: *Erigeron heleniae* G.L. Nesom; Boraginaceae: *Cryptantha geohintonii* B.L. Turner; Caryophyllaceae: *Drymaria pattersonii* B.L. Turner; Ehretiaceae: *Tiquilia tuberculata* A.T. Richardson; Polemoniaceae: *Giliastrum gypsophilum* (B.L. Turner) J.M. Porter; Solanaceae: *Chamaesaracha geohintonii* Averett and B.L. Turner. |
| 3<br>Galeana-Aramberri-Zaragoza | Asparagaceae: *Jaimehintonia gypsophila* B.L. Turner; Asteraceae: *Ageratina gypsophila* B.L. Turner, *Erigeron gypsoverus* G.L. Nesom, *E. pattersonii* G.L. Nesom, *Helianthella gypsophila* B.L. Turner, *Heterotheca gypsophila* B.L. Turner, *Hieracium gypsophilum* B.L. Turner, *Isocoma gypsophila* B.L. Turner, *Sanrobertia gypsophila* B.L. Turner, *Tridax hintoniorum* B.L. Turner, *Verbesina aramberrana* B.L. Turner, *V. hintoniorum* B.L. Turner, *V. zaragosana* B.L. Turner; Boraginaceae: *Antiphytum hintoniorum* L.C. Higgins and B.L. Turner, *Cryptantha gypsites* I.M. Johnst.; Cactaceae: *Aztekium hintonii* Glass and Fitz Maurice, *Geohintonia mexicana* Glass and Fitz Maurice, *Rapicactus beguinii* subsp. *hintoniorum* (A.Hofer) Lüthy, *R. booleanus* (G.S.Hinton) D.Donati, *R. zaragozae* (Glass and R.A.Foster) D.Donati, *Turbinicarpus hoferi* Lüthy and A.B. Lau; Campanulaceae: *Calcaratolobelia margarita* (E. Wimm.) Wilbur, *C. pringlei* (B.L. Rob.) Wilbur; Caryophyllaceae: *Arenaria hintoniorum* B.L. Turner; Commelinaceae: *Callisia hintoniorum* B.L. Turner, *Gibasis gypsophila* B.L. Turner; Crassulaceae: *Sedum booleanum* B.L. Turner; Cyperaceae: *Carex gypsophila* Reznicek and S. González; Euphorbiaceae: *Euphorbia correlli* M.C. Johnst., *E. neilmulleri* M.C. Johnst.; Fabaceae: *Dalea gypsophila* Barneby, *Dermatophyllum juanhintonianum* (B.L. Turner) B.L. Turner; Frankeniaceae: *Frankenia margaritae* Medrano; Hydrophyllaceae: *Phacelia hintoniorum* B.L. Turner, *P. vossii* N.D. Atwood; Iridaceae: *Sisyrinchium microbracteatum* G.L. Nesom; Lamiaceae: *Hedeoma ciliolata* (Epling) R.S. Irving, *H. pusilla* (R.S. Irving) R.S. Irving, *Salvia gypsophila* B.L. Turner, *Scutellaria hintoniorum* Henrickson, *S. lutilabia* T.M. Lane and G.L. Nesom; Lentibulariaceae: *Pinguicula immaculata* Zamudio and Lux, *P. nivalis* Luhrs and Lampard, *P. rotundiflora* Studnička; Liliaceae: *Calochortus marcellae* G.L. Nesom; Linaceae: *Linum gypsogenium* G.L. Nesom, *L. modestum* C.M. Rogers; Namaceae: *Nama hitchcockii* J.D. Bacon; Oleaceae: *Menodora gypsophila* B.L. Turner, *M. hintoniorum* B.L. Turner; Onagraceae: *Oenothera stubbei* W. Dietr., P.H. Raven and W.L. Wagner; Orobanchaceae: *Agalinis gypsophila* B.L. Turner, *Castilleja galehintoniae* G.L. Nesom; Papaveraceae: *Hunnemannia hintoniorum* G.L. Nesom; Poaceae: *Muhlenbergia jaime-hintonii* P.M. Peterson and Valdés-Reyna; Polygalaceae: *Polygala oedophylla* S.F. Blake; Polygonaceae: *Eriogonum fimbriatum* W.J. Hess and Reveal; Pteridaceae: *Gaga hintoniorum* (Mendenh. and G.L. Nesom) F.W. Li and Windham; Rubiaceae: *Galium dempsterae* B.L. Turner, *G. juniperinum* Standl.; Scrophulariaceae: *Leucophyllum alejandrae* G.L. Nesom, *L. hintoniorum* G.L. Nesom |
| 4<br>San Luis Potosí | Asteraceae: *Neonesomia johnstonii* (G.L. Nesom) Urbatsch and R.P. Roberts, *Tridax candidissima* A. Gray, *Verbesina potosina* B.L. Rob.; Iridaceae: *Sisyrinchium zamudioi* Espejo, López-Ferr. and Ceja; Lentibulariaceae: *Pinguicula gypsicola* Brandegee, *P. takakii* Zamudio and Rzed.; Liliaceae: *Calochortus mendozae* Espejo, López-Ferr. and Ceja; Linaceae: *Linum macradenium* Brandegee |
| 5<br>Colima | Crassulaceae: *Graptopetalum glassii* Acev.-Rosas and Cházaro; Lentibulariaceae: *Pinguicula colimensis* McVaugh and Mickel |
| 6<br>Oaxaca-Mixteca | Bromeliaceae: *Hechtia gypsophila* López-Ferr., Espejo and Hern.-Cárdenas; Crassulaceae: *Echeveria subcorymbosa* Kimnach and Moran; Lentibulariaceae: *Pinguicula pygmaea* Rivadavia, E.L.Read and A.Fleischm. |
| 7<br>Oaxaca-Teita | Asparagaceae: *Xochiquetzallia magnifolia* García-Mend. and J.Gut.; Asteraceae: *Mixtecalia teitaensis* Redonda-Mart., García-Mend. and D. Sandoval; Cactaceae: *Cephalocereus parvispinus* S. Arias, H. J. Tapia and U. Guzmán |

*3.4. Conservation Status of the Mexican Gypsicolous Flora*

The geoprocessing methodology identified nine federal PNAs that contain gypsum soil areas in Mexico (Figure 5a–f). Two of them were located in the Baja California Peninsula (a, b) and corresponded to the "Área de Protección de Flora y Fauna (APFF) Valle de Los

Cirios" and "APFF Islas del Golfo de California". In Chihuahua, one gypsum locality occurred at the "APFF Cañón de Santa Elena" (c). The gypsisols of the Cuatro Ciénegas Basin and the adjacent areas are protected by two PNAs: the "APFF Cuatro Ciénegas" and the "Área de Protección de los Recursos Naturales Cuenca Alimentadora del Distrito Nacional de Riego (CADNR) 4 Don Martín" (d). Within the limits of Coahuila and Nuevo León, we found that the "Parque Nacional Cumbres de Monterrey" and the "CADNR 26 Bajo Río San Juan" hosted gypsum areas (e). Finally, we identified two PNAs that have gypsum outcroppings in Oaxaca (f): the "APFF Boquerón de Tonalá", in the Mixteca region, and the "Reserva de la Biosfera Tehuacán-Cuicatlán", which is shared with Puebla.

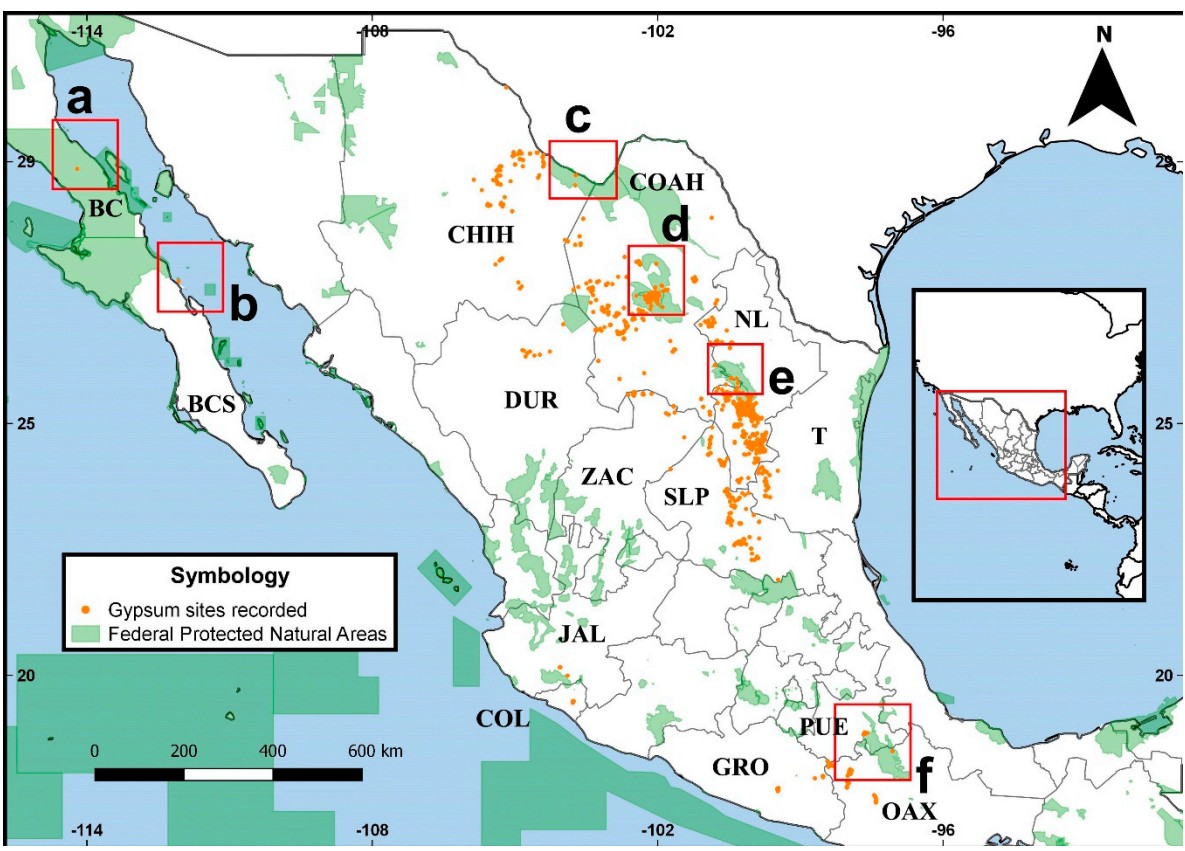

**Figure 5.** Protected Natural Areas (PNAs) of Mexico with gypsum records. (**a**) Área de Protección de Flora y Fauna (APFF) Valle de Los Cirios, (**b**) APFF Islas del Golfo de California, (**c**) APFF Cañón de Santa Elena, (**d**) APFF Cuatro Ciénegas and Área de Protección de los Recursos Naturales Cuenca Alimentadora del Distrito Nacional de Riego (CADNR) 4 Don Martín, (**e**) Parque Nacional Cumbres de Monterrey and CADNR 26 Bajo Río San Juan, and (**f**) APFF Boquerón de Tonalá and Reserva de la Biosfera Tehuacán-Cuicatlán.

Of the 1470 species recorded growing on gypsum, 56 are included under the NOM-059-SEMARNAT-2010. Six of these are catalogued as endangered, twenty-two are threatened, and twenty-eight are subject to special protection. Among these fifty-six catalogued species, only six were classified as gypsophytes.

## 4. Discussion

### 4.1. Botanical Records on Mexican Gypsum Soils

Gypsum sites are scattered throughout Mexico. Even though we recorded references to gypsum sites in 14 states of the country, we are aware that there are many unexplored gypsum outcrops that we failed to detect with our workflow. For example, the SGM [9] reports the presence of gypsum soil in Campeche, Hidalgo, Michoacán, Morelos, Sinaloa,

and Sonora, but we did not find botanical records for gypsum in these states. Martínez and Galindo-Leal [15] reported plants growing on gypsum in the state of Campeche, but we were unable to locate herbarium collections. There are numerous botanical collections on gypsum in Mexico whose collecting labels do not identify the collecting site as possessing gypsisols. An important fact is that gypsum is often confused with calcium carbonate or saline soils [73]. To corroborate this, different approaches such as satellite remote sensing, geological surveys, soil samplings, and ecological data should be used in an integrative way. These are complicated and time-consuming tasks in a country as large as Mexico, especially considering information is often lacking (e.g., soil sampling and ecological data). Additional work is necessary to complete the gypsicolous flora of Mexico and to corroborate if the presence of gypsum reported by the SGM [9] represents surface outcrops and have direct contact with plants or if the deposits are located in mines. Nevertheless, our results are an important first step to detect potential gypsum outcrops in Mexico and to understand their flora.

For much of the botanical history of Mexico, only the gypsophile flora from the CHIH have received significant botanical research interest [5,8,10,13,38]. However, we identified that, with the exception of the Cuatro Ciénegas Basin [34,44], most of the gypsum outcrops in CHIH have poor botanical records (Figure 2b). This is particularly evident in the region of west central Coahuila, which has some of the most extensive and least accessible gypsum exposures in North America, as part of the Acatita Formation [44]. The flora associated with the gypsum outcrops from the SMOr seems to be better known, mostly due to the fieldwork conducted in Nuevo León by the Hinton family [41] and in San Luis Potosí by Rzedowski [74]. However, our results showed that many areas of the SMOr need more intensive exploration, even in the relatively well-explored states of Nuevo León and San Luis Potosí. For example, San Luis Potosí has two gypsiferous regions, one in the Middle Zone that occupies 203,950 Ha and another in the Highland Zone which has 193,309 Ha [75], but the flora of the Middle Zone is less studied.

Meanwhile, we found very few records from the outcrops of Baja California, Baja California Sur, Colima, Durango, Jalisco, Puebla, and Zacatecas. With the exception of one recent floristic inventory in Colima [17], research on these gypsum outcrops is scarce and focused on particular groups [16,76,77]. On the other hand, the recent exploration of some gypsum communities in Oaxaca has revealed a highly localized endemic flora [78]. Perhaps the most outstanding example is the genus *Mixtecalia*, a tree-like Asteraceae [21]. Due to the lack of collections and the particularities of gypsisols in Mexico, we are sure that the intensification of botanical exploration in these communities will lead to the description of many new taxa.

### 4.2. Species Richness

The highest total species richness was documented in the Cuatro Ciénegas Basin, which is the most sampled gypsum area of the country and has been previously recognized as a region of outstanding diversity [34,44,79]. Additionally, the area surrounding Santo Domingo Tonalá in Oaxaca hosts an elevated number of species (Figure 3a). This zone has received extensive attention, with many botanical collections and technical studies [80,81]. It is situated in the Balsas Basin, a biogeographic province with tropical affinity and with wetter conditions than the gypsisols of northern Mexico. In tropical gypsisols, the rain washes the surface of the soil and hinders the formation of a hard gypsum crust, which allows the establishment of a greater proportion of gypsovag species [3,7], but comparatively fewer gypsum endemics. Nevertheless, Oaxaca hosts the highest species richness of angiosperms in Mexico [82,83], which contributes to the diversity of plants on gypsum outcrops there.

In Mexico, the highest species richness levels of vascular plants are concentrated in the southern portion of the country [84], following the global latitudinal pattern [52]. It is feasible that the gypsicolous floras of Mexico as a whole follow this pattern too. Total species richness was highly correlated with the number of occurrence records; thus, it

is possible that if the number of records in southern and western Mexico increases, the number of species may be higher than in those in northern Mexico. The pattern of spatial distribution of the total species richness in the country will change as the poorest explored areas increase their records, particularly those from southern and western Mexico. However, this pattern is different when species richness is analyzed only with gypsophytes.

Gypsophyte species richness is concentrated in northern Mexico, mainly in the SMOr, but also in the CHIH (Figure 3b). Gypsum soils are widely distributed in these regions, with individual outcrops, the extent of which varies in size from many square kilometers to a few square meters [6,8,9]. It is believed that gypsophyte floras evolved from groups that have their diversity centers in the same regions [6]. Within the SMOr and CHIH, the combination of isolation of individual outcrops from one another, a wide range of different climatic conditions, and the relative climatic stability of the gypsum outcrops throughout the Pleistocene have promoted the evolution of gypsophyte plant lineages that occur there [6,8,38,85]. Several studies have demonstrated the existence of gypsophyte clades in the CHIH and SMOr (with as many as 10 gypsophyte species in a single clade), implying that speciation occurred after the shift to a gypsophyte ecology [13,86–90]. In the case of the SMOr, the large topographical and climatic variation has promoted plant diversification [51,91,92]. Our results follow this pattern also for the gypsum outcrops within this biogeographic province at Nuevo León. Within the CHIH, the Cuatrociénegas Basin hosts outstanding levels of diversity [79]. Most of the bolson consists of surface gypsisols; thus, gypsum plays a main role in community assembly and differentiation. Here, species assemblages have been shown to be related to the effect of topographic factors, altitude, and slope [93]. The gypsophyte diversity estimated in the municipality of Cuatro Ciénegas by Ochoterena et al. [44] and this work exemplify this well.

Mexico has a megadiverse flora, which results from many factors, including its geographic location that lies at the contact of Neotropical and Nearctic elements; its climatic history, comprising of periods of fluctuations and stability; and a highly varied topography, physiography, and geology [57,94]. The total number of plant species in the country is estimated to be 22,969 [95]. Our work reports that 1470 (6.4%) of those species grow on gypsum, from which 205 (14% of the gypsicolous plants in Mexico) are gypsophytes. Hence, gypsophytes represent almost 1% of the total flora of the country. This is the first work to attempt such approximations; thus, we are aware that these numbers are far from definitive. On the one hand, it is possible that some of the putative gypsophytes are simply yet to be recorded outside gypsum. On the other hand, in-depth taxonomic and phylogenetic studies, as well as botanical exploration of the gypsum outcrops in Mexico, are highly incomplete, and the discovery of additional gypsophytes (perhaps many more) is to be expected. Regardless, Mexico hosts a highly species-rich gypsophyte flora, considering that the first preliminary estimate of the number of gypsophytes across all of the Palearctic and Australia resulted in 378 species, from which Iran has the highest number, with 91 [96]. Consequently, given the current knowledge, our work places Mexico as the country with the highest recorded number of gypsophyte species worldwide. A detailed taxonomic analysis of the gypsophyte flora of Mexico is in preparation.

*4.3. Endemism*

The use of the CWE index allowed us to identify the centers of endemism with the highest numbers of localized gypsophyte species. Our results recovered seven such centers of endemism across Mexico (Figure 4a). Center 1, the Cuatro Ciénegas basin, has been recognized as an important center of endemism and biodiversity for many organisms and is home to many narrowly endemic gypsophytes [11,34,44,79]. Centers 2 and 3, in the SMOr, have been identified as angiosperm diversity hotspots [50,51,97]. The high number of endemic species that occur in these zones is remarkable [37,56]. Center 4, which includes the gypsophilous grasslands in the CHIH and a small portion of the SMOr, with xerophilous scrubland in Guadálcazar, San Luis Potosí, hosts an important level of endemism [50,51]. Center 5, in western Mexico at the Microcuenca La Salada in Colima, is another area with

recognized high diversity and endemism [17]. Despite the geographic proximity of the two nearby centers of endemism in Oaxaca (6 and 7), they have very different species compositions. Center 6 is composed mainly of species characteristic of tropical deciduous forests, while center 7 is composed of xerophilous scrubland. Even though these sites do not possess high gypsophyte richness, those that occur are microendemic. Overall, the gypsum outcrops of Mexico contain elevated rates of endemism, with those in the CHIH and the SMOr hosting many endemic gypsophyte taxa, but those in western and southern Mexico uniformly hosting very rare gypsophyte species (Figure 6).

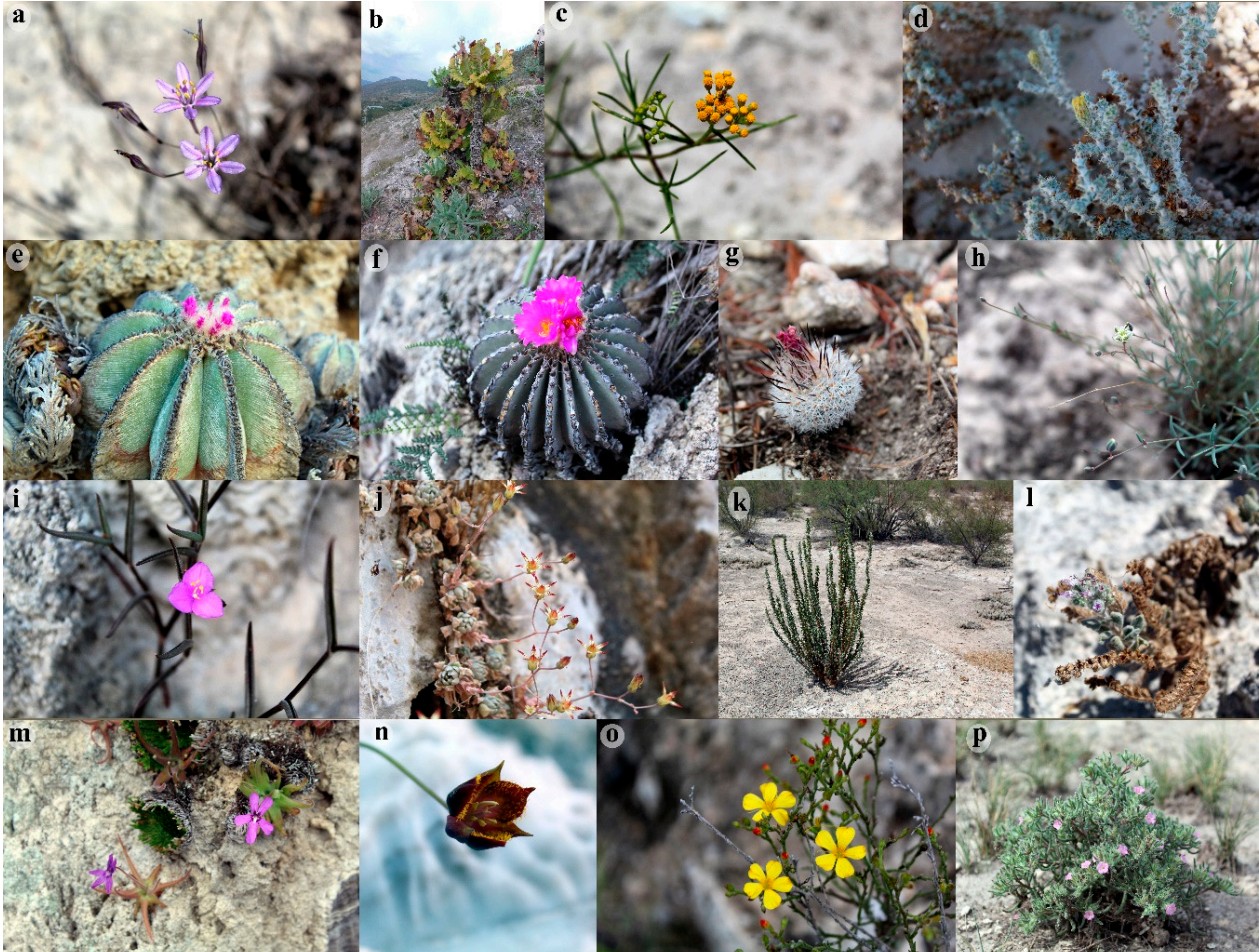

**Figure 6.** Examples of gypsophytes of Mexico. Asparagaceae: (**a**) *Jaimehintonia gypsophila* B.L. Turner; Asteraceae: (**b**) *Mixtecalia teitaensis* Redonda-Mart., García-Mend. and D. Sandoval, (**c**) *Sartwellia mexicana* A. Gray, (**d**) *Xanthisma restiforme* (B.L. Turner) D.R. Morgan and R.L. Hartm.; Cactaceae: (**e**) *Aztekium hintonii* Glass and Fitz Maurice, (**f**) *Geohintonia mexicana* Glass and Fitz Maurice, (**g**) *Rapicactus booleanus* (G.S.Hinton) D.Donati; Caryophyllaceae: (**h**) *Drymaria lyropetala* I.M. Johnst.; Commmelinaceae: (**i**) *Callisia hintoniorum* B.L. Turner; Crassulaceae: (**j**) *Graptopetalum glassii* Acev.-Rosas and Cházaro; Fouquieriaceae: (**k**) *Fouquieria shrevei* I.M. Johnst.; Hydrophyllaceae: (**l**) *Phacelia marshall-johnstonii* N.D. Atwood and Pinkava; Lentibulariaceae: (**m**) *Pinguicula gypsicola* Brandegee; Liliaceae: (**n**) *Calochortus marcellae* G.L. Nesom; Linaceae: (**o**) *Linum macradenium* Brandegee; Namaceae: (**p**) *Nama stevensii* C.L. Hitchc. Photographs: (**a,c–p**) by J. P. Ortiz-Brunel, (**b**) by Juvenal Aragón.

### 4.4. Conservation Status of the Mexican Gypsicolous Flora

Nine federal PNAs currently contain gypsum outcrops (Figure 5). Unfortunately, only the "APFF Cuatro Ciénegas" and the "CADNR 4 Don Martín" occur in a zone (Cuatro Ciénegas Basin) with high gypsophyte richness and endemism. Even though the "Parque

Nacional Cumbres de Monterrey" and the "CADNR 26 Bajo Río San Juan" host some gypsum outcrops, the highest concentration of records, species richness, and endemism of gypsophytes in the SMOr occurs just south of them. This problem has also been identified for the endemic vascular plants of the SMOr [51]. If the "CADNR 26 Bajo Río San Juan" in Nuevo León were enlarged, it would cover the aforementioned gypsum areas. Additionally, this would protect other important ecosystems in one of the most relevant hotspots of endemism and diversity in Mexico [50,51]. Meanwhile, some gypsum outcrops in Chihuahua and Oaxaca with few gypsophyte taxa are protected within the "APFF Cañón de Santa Elena" and the "Reserva de la Biósfera Tehuacán-Cuicatlán", respectively. Fortunately, one of the areas with the highest total species richness is protected by the "APFF Boquerón de Tonalá" in Oaxaca.

Despite the existence of the above protected areas, our results show that most of the gypsum plant communities with the highest species richness and endemism of the gypsophyte flora of Mexico are unprotected. This problem is coupled with the fact that only six gypsophytes are officially protected by Mexican law [72], despite the fact that most gypsophytes have a very narrow geographic distribution and grow on a substrate frequently exploited by mining. Gypsum mining represents one of the most significant threats to gypsum biodiversity worldwide [3], and those of Mexico are no exception. Gypsum is not abundant in much of the country; thus, exposed outcrops are often economically exploited [98], and it is important that this activity is regulated based on species diversity and endemism. Finally, new PNAs should be developed to promote the conservation of these unique communities, particularly in zones with high species richness and endemism.

## 5. Conclusions

Our work represents the first attempt to integrate information about plant species growing on gypsum outcrops throughout Mexico based on botanical collections. We find that most of the potential gypsum outcrops in Mexico are poorly explored and deserve more attention, as in general, the gypsicolous flora of Mexico is diverse and unique. Even though the gypsophyte flora of Mexico is not completely known, our work demonstrates that it is the most species-rich in the world, and that nearly 1% of Mexican plant species are gypsophytes. The botanical exploration of gypsum communities is likely to result in the discovery of new gypsophytes and in an increase in the total number of species of the gypsicolous flora in the country. This seems particularly likely and relevant for those exposures in western and southern Mexico, which largely occur in tropical seasonal forests rather than in the more typical arid and semi-arid vegetation. Despite the high diversity and endemism of gypsum outcrops in Mexico, those with higher numbers of gypsophytes and those that contain narrowly endemic species are unprotected, with the exception of Cuatro Ciénegas. We found that enlarging the PNA CADNR 26 Bajo Río San Juan in Nuevo León would protect one of the most important hotspots of endemism in Mexico. Additionally, most gypsophytes need to be evaluated by IUCN criteria to assess their conservation status. We hope that this work will encourage the botanical exploration of gypsum outcrops in Mexico, and will set a necessary baseline for future ecological, taxonomic, and evolutionary research to understand these communities and conserve their biodiversity.

**Author Contributions:** Conceptualization, J.P.O.-B., H.O., M.J.M. and H.F.-O.; data curation, J.P.O.-B., H.O., M.J.M., J.A.-P., G.M.-L., A.R., M.M.S.-R. and H.F.-O.; formal analysis, J.P.O.-B.; investigation, J.P.O.-B., J.F., G.M.-L., A.R., M.M.S.-R. and H.F.-O.; methodology, J.P.O.-B., H.O., M.J.M. and H.F.-O.; resources, J.F.; visualization, J.A.-P. and J.F.; writing—original draft, J.P.O.-B. and H.F.-O.; writing—review and editing, J.P.O.-B., H.O., M.J.M., J.A.-P., J.F., G.M.-L., A.R., M.M.S.-R. and H.F.-O. All authors have read and agreed to the published version of the manuscript.

**Funding:** This research received no external funding.

**Data Availability Statement:** The database with the botanical records of vascular plants on gypsum is free and available upon request.

**Acknowledgments:** We thank George Yatskievych and the Billie L. Turner Plant Resources Center of the University of Texas at Austin for their invaluable help. We are grateful to Arantzazu L. Luzuriaga for their help, comments, and suggestions. We thank Rosa de Lourdes Romo Campos for her technical support. The authors thank the Gypworld project for promoting collaboration and ideas interchange.

**Conflicts of Interest:** The authors have no competing interest to declare that are relevant to the content of this article.

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
