# Peer review of "Patterns of Richness and Endemism in the Gypsicolous Flora of Mexico"

_diversity, doi:10.3390/d15040522_

Round 1
Reviewer 1 Report
The manuscript with ID: diversity-2286830, entitled “Patterns of richness and endemism in the gypsicolous flora of Mexico” by Juan Pablo Ortiz-Brunel and co-workers presents a comprehensive analysis of the botanical records reported on gypsum soils in Mexico. The study will make a highly valuable contribution to our understanding of the gypsophylic flora of America and the world, being Mexico the region with a highest reported number of gypsophilic taxa of the world. The study seems to have been thoroughly conducted and the manuscript is very well written. I just have some concerns that I think can be easily addressed by the authors by changing the main focus of the manuscript and introducing some caveats in the interpretation of part of their results. I explain these below:
- As it is now written, the focus of the paper is often set on the soils (e.g. Page 7, 1st paragraph, see also other comments in the annotated pdf). However, no soil analyses were made as part of this research. The documentation of gypsisols is made through the identification of plants that were supposedly collected on such soils, but no back up analytical results are provided to support such approach (which may fall into a certain circularity). As the authors acknowledge in the discussion of the manuscript, this approach is risky. Throughout my experience surveying gypsum outcrops across the world I have very often looked for citations of plants supposedly collected on gypsum and found they were collected on carbonates or limestones. This happened in almost every gypsum region of the world I have worked on. The assumption that the description of soils included in herbarium labels is correct has, hence, some risk associated. To overcome these limitations, and provided the main actual focus of the study are not the gypsum soils but the plants that grow on them, I would suggest shifting that focus throughout the text, particularly in the introduction (last paragraph), materials and methods and results (you can see my comments in the annotated pdf for guidance). This will clear out many doubts on the procedure followed: the aim was not to look for gypsum soils, but to map/evaluate/study gypsophilic plant records.
In relation to this, it would be nice to see up to what point gypsophilic records and actual gypsum soil records match (according to geological surveys available). Currently only the mismatch between both types of records is shown.
- One of the analyses made is the estimation of species richness for certain sites. This estimation is based on botanical records and not on actual vegetation releves/plots/transects. This approach is very risky and I would suggest removing it from the manuscript or introducing caveats on the analysis of these results. For most of the sites, botanical records may not be representative of the whole flora. Simply because common species or taxa found of low interest may not be reported in such approaches. Only records that were collected with the purpose of capturing plant communities as a whole should be included and, I am afraid, this was not the case for most of the data used. The risk of the approach followed is that species richness can be largely underestimated. Further, the differences in the species richness estimated may just be a result of the differential sampling effort, as the authors report in their results.
I have some other minor issues that I have highlighted in the annotated pdf. I hope my comments help the authors get the best out of their excellent work.

Reviewer 2 Report
This is a well-presented, well-designed and interesting paper on an important topic, given that Mexico has the most diverse gypsophyte flora in the world despite being only partially documented and that the regions with the highest species richness and endemism are not protected. The Figures are good and the findings have broad relevance for promoting the study and conservation of this flora in Mexico.
More detail on the literature review is needed in the Methods (currently this is only one sentence). For example, what were the key search terms? What sorts of literature were interrogated? This is particularly important given that numerous gypsum sites were not detected using the methods described in this study. It seems that a full literature review was not the focus of this study, but rather a review of herbarium collections.
Was remote sensing used to identify gypsisol sites? This is a good way of detecting gypsisol soils in many areas due to their distinctive appearance on satellite imagery. Such data are widely and freely available and should be incorporated into this study.
Table 1 is very long - is it necessary to include the authorities for each scientific name?
Incorporating satellite imagery analysis into this review would in my opinion greatly enhance its reach and scope, thus this represents a moderate level of revision to the manuscript. Alternatively, the authors should justify why this approach was not taken.
Reviewer 3 Report
Keywords
One of the most used words in SCOPUS and other databases in articles on flora associated with gypsum outcrops is “gypsophile”, used by some authors as an adjective and by others as a noun. Since these words are neither in the title of the article nor in the abstract, I recommend that the last sentence of the abstract be written like this: Mexico hosts the most diverse gypsophile flora in the world despite having been partially studied and collected. The regions with the highest species richness and endemism are unprotected. Authors can see how database searches using "gypsophyte" or "gypsophile" yield different, although partially overlapping, results.
Also in the keywords I would change "diversity" to "biodiversity"
Materials and methods
Mexican gypsum soils
Although throughout the main text of the manuscript gypsum soils (gypsisols) are made almost synonymous with areas with gypsum outcrops, I believe that the heading "Mexican gypsum soils" should be titled “Mexican gypsum areas”. Perhaps here too the authors may want to consider the possibility of including some summary information on the gypsiferous soils of Mexico, especially in relation to their gypsum content (assuming that the data are available). Regarding the climate, two or three bioclimatic diagrams could help potential readers to better understand the desert nature of the vast territory studied and the seasonality of rainfall. Another possibility is to refer the reader to the bibliographical sources in which this information can be consulted.
The heading "Conservation in gypsum communities" would be more informative if it were named Protected areas and gypsum flora (or gypsophile flora) conservation.
In addition to table 1, have the authors considered the possibility of including in an appendix the 205 species of gypsophytes found? I am aware that, very generously, they offer it upon request, but I believe that the information is so valuable that it deserves to be part of the manuscript, at least as an appendix, but I understand that this is a decision that is the responsibility of the authors.
Let me make an observation: if you change the headings in the M&M section, as I have suggested, you should also do the same in the results and discussion.
Finally, I would like to point out that, although the article is acceptable in its current form, I have greatly missed references to the protection of gypsophile flora in a direct way. I am referring to its inclusion in threat catalogs or red lists. I have not found that these lists or red books are available for the vascular flora in Mexico, but I do know that this legal document exists: 2002. Norma Oficial Mexicana NOM-059-SEMARNAT-2001 Protección ambiental–especies nativas de México de flora y fauna silvestres. (https://www.dof.gob.mx/normasOficiales/4254/semarnat/semarnat.htm)
